# Organ Utilization Rates from Non-Ideal Donors for Solid Organ Transplant in the United States

**DOI:** 10.3390/jcm13113271

**Published:** 2024-05-31

**Authors:** Steven A. Wisel, Daniel Borja-Cacho, Dominick Megna, Michie Adjei, Irene K. Kim, Justin A. Steggerda

**Affiliations:** 1Department of Surgery, Cedars-Sinai Medical Center, Los Angeles, CA 90048, USA; steven.wisel@cshs.org (S.A.W.); michie.adjei@cshs.org (M.A.); irene.kim@cshs.org (I.K.K.); 2Comprehensive Transplant Center, Cedars-Sinai Medical Center, Los Angeles, CA 90048, USA; 3Division of Transplant Surgery, Department of Surgery, Northwestern Memorial Hospital, Chicago, IL 60611, USA; 4Division of Cardiothoracic Surgery, Smidt Heart Institute, Cedars-Sinai Medical Center, Los Angeles, CA 90048, USA

**Keywords:** organ transplant, organ procurement, donation after brain death, donation after circulatory death, donor pool, organ discard

## Abstract

**Background:** Non-ideal donors provide acceptable allografts and may expand the donor pool. This study evaluates donor utilization across solid organs over 15-years in the United States. **Methods:** We analyzed the OPTN STAR database to identify potential donors across three donor eras: 2005–2009, 2010–2014, and 2015–2019. Donors were analyzed by a composite Donor Utilization Score (DUS), comprised of donor age and comorbidities. Outcomes of interest were overall and organ-specific donor utilization. Descriptive analyses and multivariable logistic regression modeling were performed. *p*-values < 0.01 considered significant. **Results:** Of 132,465 donors, 32,710 (24.7%) were identified as non-ideal donors (NID), based on a DUS ≥ 3. Compared to ideal donors (ID), NID were older (median 56 years, IQR 51–64 years vs. 35 years, 22–48 years, *p* < 0.001) and more frequently female (44.3% vs. 39.1%, *p* < 0.001), Black (22.1% vs. 14.6%, *p* < 0.001) and obese (60.7% vs. 19.6%, *p* < 0.001). The likelihood of overall DBD utilization from NID increased from Era 1 to Era 2 (OR 1.227, 95% CI 1.123–1.341, *p* < 0.001) and Era 3 (OR 1.504, 1.376–1.643, *p* < 0.001), while DCD donor utilization in NID was not statistically different across Eras. Compared to Era 1, the likelihood of DBD utilization from NID for kidney transplantation was lower in Era 2 (OR 0.882, 0.822–0.946) and Era 3 (OR 0.938, 0.876–1.004, *p* = 0.002). The likelihood of NID utilization increased in Era 3 compared to Era 1 for livers (OR 1.511, 1.411–1.618, *p* < 0.001), hearts (OR 1.623, 1.415–1.862, *p* < 0.001), and lungs (OR 2.251, 2.011–2.520, *p* < 0.001). **Conclusions:** Using a universal definition of NID across organs, NID donor utilization is increasing; however, use of DUS may improve resource utilization in identifying donors at highest likelihood for multi-organ donation.

## 1. Introduction

The number of transplants performed annually in the United States has been increasing since the turn of the century (UNOS/OPTN data, https://optn.transplant.hrsa.gov/data/view-data-reports/request-data/, accessed on 3 December 2022). Despite this, there is an ongoing disparity between the number of patients awaiting transplant and the availability of acceptable donor organs. The utilization of marginal donors has been promoted to meet this persistent need.

The definition of marginal donors varies across organs and there is no single consensus definition. Kidney donors may be deemed expanded criteria donors (ECD), defined as brain-dead donors (DBD) ≥60 years old or 50–59 years old with at least two of the following features: history of hypertension, terminal serum creatinine (SCr) > 1.5 mg/dL, or cerebrovascular disease as primary cause of death (COD) [1]. This definition notably does not consider kidney donor risk index (KDRI) score [2]. donation after circulatory death (DCD) donor status, or other comorbid conditions, such as diabetes or obesity, which have been associated with the development of chronic kidney disease [3]. For other organs, the definitions of “marginal” are vague. Feng et al. created the Donor Risk Index (DRI) in 2006 for liver allografts, which associated increased risk of graft loss with donor age >40 years, DCD donors, split/partial grafts, Black donors, shorter donors, and cerebrovascular accident as COD [4]. Others have considered graft macrosteatosis above 30%, donor age >70 years, national share grafts, prolonged cold ischemic time (CIT) and/or warm ischemic time (WIT), hepatitis C virus (HCV) positive donors, and high inotrope requirement as factors making donors marginal [5,6,7,8,9]. Heart and lung donors are evaluated more strictly. The ideal lung donor is typically 25–40 years old, has no smoking history, and follows clinical parameters including a clear chest X-ray, PaO_2_/FiO_2_ ratio <350, and normal bronchoscopy. Heart donors may be considered marginal with age >50 years old, ejection fraction <50%, left ventricular hypertrophy, cocaine use, obesity, race mismatch, gender mismatch, and elevated blood urea nitrogen/SCr ratio [10,11,12].

Over the past two decades, the general population of the United States (U.S.) has seen significant increases in the prevalence of chronic comorbid conditions, including obesity, diabetes, and hypertension, amongst others [13]. As the age and health of the general population in the U.S. changes, an increasing number of donors will meet existing marginal donor criteria for at least one organ. This rise in comorbidities, coupled with the increased overall volume of potential donors undergoing evaluation, raises the complexity of identifying donors that are likely to proceed to multi-organ donation. The implications of futile donor evaluations produces strain on the transplant system, including organ procurement organization (OPO) staff, transplant coordinators, physicians, and surgeons. Here, we sought to identify trends in overall and organ-specific donor utilization using a global definition of Non-Ideal Donor to identify donor demographics most likely to result in multi-organ donation. Of note, we have selected the terms Ideal Donor (ID) and Non-Ideal Donor (NID) to avoid confusion with existing definitions, including “marginal” and “ECD” donors. 

## 2. Materials and Methods

### 2.1. Patient Population

A retrospective cohort study was performed with the Organ Procurement and Transplant Network Standard Transplant Analysis and Research (STAR) database, which was reviewed for all deceased donors in the U.S. between 1 January 2005, and 31 December 2019. Variables selected for analysis included demographic characteristics: age, gender, ethnicity, height, weight, body mass index (BMI), COD, and donation after brain death (DBD) or donation after circulatory death (DCD) status; medical/social characteristics such as history of diabetes, hypertension, hepatitis C virus (HCV) or hepatitis B virus (HBV) positivity, any drug use, cigarette smoker (>20 pack-year history), heavy alcohol use, or otherwise meeting Center for Disease Control (CDC) High Risk criteria. Kidney and liver biopsies were not uniformly performed and therefore were not included in analysis or considered as variables for NID. 

The definition of a NID used herein was derived from evaluation of current organ-specific definitions of extended criteria donors (Table 1). Based on factors common across organ-specific definitions, or markers of overall donor comorbidity, we developed a Donor Utilization Score (DUS) that was determined by a combination comorbid conditions: any donor >70 years old (3 points), donors 50–69 years old (1 point) or any of the following: BMI >30 kg/m^2^, diabetes, hypertension, prior MI, HCV-positive, and terminal SCr >1.5 mg/dL (1 point each, Table 2). To reduce confounding, we separated DBD and DCD cohorts for pertinent portions of the analysis. While additional variables may influence organ-specific utilization, those which were inconsistently reported, such as donor inotrope use, echocardiographic findings, and arterial blood gas analyses, were excluded from the definition. 

Donor utilization was determined by transplantation of at least one organ. Multi-organ donors were defined as having at least two different organ types transplanted (e.g., liver and kidney) into one or more recipients; however, a donor of two lungs or two kidneys alone was not considered a multi-organ donor. 

The primary aim of this study is to evaluate changes in the utilization of NID over time. To properly assess changes over time, the study period was divided into different eras. Given that allocation policies changed for different organs at different times (i.e., SHARE 35 for liver allocation in 2013, Kidney Allocation System in 2015 and 2018, heart allocation changes in 2018, and Lung Allocation Score in 2005), the study period was divided into three separate eras of equal duration: Era 1 from 2005–2009, Era 2 from 2010–2014, and Era 3 from 2015–2019. Donor characteristics were evaluated within and across eras.

### 2.2. Logistic Regression Modeling

Univariate and multivariable logistic regression modeling were utilized to evaluate the likelihood of donor utilization. Multivariable analysis was performed with a composite model consisting of key donor demographic and medical variables: donor type (DBD or DCD), age, race, BMI, COD, blood type, history of hypertension, diabetes, prior MI, HCV-positive, HBV-positive, heavy alcohol use, cigarette smoker, any drug use, and terminal Cr >1.5 mg/dL, terminal ALT >150, and INR >1.5. The utilization of NID was evaluated across eras for all organs as well as on an organ-specific basis. Odds ratios (OR) and 95% confidence intervals (CI) are reported for all regression models.

### 2.3. Data Analysis

Variables were compared across groups using *t*-tests, chi-square, one-way analysis of variance (ANOVA), and Wilcoxon rank-sum tests, as appropriate. Changes over time were evaluated using Cochran-Armitage test of trends. A *p*-value < 0.01 was considered significant to reduce the risk of false positive findings. All analyses were performed using JMP Pro 16 software (SAS Inc., Cary, NC, USA).

## 3. Results

### 3.1. Validation of Donor Utilization Score

To assess the use of global donor characteristics to predict organ utilization, we created a Donor Utilization Score (DUS) as described above (Table 2). In an overall analysis, a rising DUS correlated with a decrease in number of potential donors, as well as a decreasing likelihood that a potential donor would lead to utilization of any organ (*p* < 0.001, Table 3). Furthermore, rising DUS corresponded to both the likelihood that a given donor would proceed to multi-organ donation (*p* < 0.001), as well as the median number of organs utilized for transplant per donor (*p* < 0.001). Using the DUS, we defined our cutoff for Non-Ideal Donors (NID) as any donor with a DUS ≥ 3. This corresponded to all donors with <50% likelihood for multi-organ donation, and with a median number of organs donated at or below 2.03 ± 1.32 organs.

Furthermore, we evaluated the DUS on an organ-specific basis (Figure 1). Lungs were least likely to be utilized overall, with 31.2% utilization from DUS 0 donors, and 1.7% utiliation from DUS 6 donors (*p* < 0.001). No lung allografts were utilized for donors with DUS >6. For heart transplantation, donors with DUS 0 had 52.9% heart utilization, decreasing to a rate of 0.1% at DUS 6 (*p* < 0.001). Again, zero heart allografts were utilized from donors with DUS >6. 95.4% of kidneys were utilized for transplant from DUS 0 donors, with a continual downtrend to 5.7% utilization from donors with DUS 8 (*p* < 0.001). No kidneys were used from donors with DUS 9. When evaluating liver utilization, increasing DUS score corresponded to a decreasing number of potential donors. However, liver allograft utilization from donors worked up for transplantation remained relatively stable, ranging from 67.6% to 82.8% (*p* < 0.001).

### 3.2. Population Characteristics

Between 2005 and 2019, 132,465 potential donors were identified for organ donation. Of those, 32,710 (24.7%) met our definition of NID. Demographic characteristics of non-deal and ideal donors are compared in Table 4. Compared to ideal donors (ID) donors, NID were older (median 56 years, IQR 51–64 years versus 35 years, IQR 22–48 years, *p* < 0.001), more frequently female (44.3% versus 39.1%, *p* < 0.001), Black (22.1% versus 14.6%, *p* < 0.001), and died of CVA/Stroke (56.5% versus 26.4%, *p* < 0.001). NID were also more frequently obese (60.7% versus 19.6%, *p* < 0.001) and had medical history significant for hypertension (87.8% versus 17.8%, *p* < 0.001) or diabetes (40.2% versus 2.4%, *p* < 0.001). Interestingly, ID had a higher incidences of heavy alcohol use (17.5% versus 16.8%, *p* = 0.002) and drug use (43.2% versus 29.3%, *p* < 0.001).

### 3.3. Trends in Non-Ideal Donors by Era

Over the study period, the number of potential donors increased annually from 7577 in 2005 to 11,830 in 2019 (*p* < 0.001 by Cochran-Armitage trend test). The proportion of NID remained relatively stable throughout the study period, accounting for 23.3% to 27.9% of donors per year (Figure 2A). Amongst NID, the proportion of DCD donors increased significantly over time from 6% in 2005 to 18% in 2019 (*p* < 0.001, Figure 2B). NID were utilized ~10% less frequently than ID amongst DBD donors (Figure 2C) and ~20% less frequently for DCD donors (Figure 2D). Across the three eras, there was a small increase in the proportion of NID utilized for transplant from 83.0% in Era 1 to 86.2% in Era 3 (*p* < 0.001, Table 5). Comparatively, the utilization of ideal donors was 97.0% to 96.7% over the same period.

### 3.4. Likelihood of Non-Ideal Donor Utilization

The likelihood of NID utilization for organ transplant varied significantly over the study period. Multivariable analysis showed that compared to Era 1, DBD organs from NID donors were more likely to be utilized for transplant of at least one organ in both Era 2 (OR 1.227 95% CI 1.123–1.341, *p* < 0.001) and Era 3 (OR 1.504, 95% CI 1.376–1.643, *p* < 0.001; Figure 3). The utilization of NID for DCD donation did not change significantly over the study period, however it showed a trend towards decreased likelihood of donor utilization.

### 3.5. Organ-Specific Donor Utilization—Kidney

The utilization of specific organs was evaluated and compared across eras. Furthermore, organ-specific donor utilization was compared between NID and ID for both DBD and DCD donors over time (Figure 4).

Kidney allografts were utilized from 44.1% of NID donors in Era 1, 46.0% in Era 2, and 46.2% in Era 3. Kidney discard rates slightly increased over time from 30.1% in Era 1 to 31.7% in Era 3 (*p* < 0.001). The utilization rates for kidney allografts from ID was similar for both DBD and DCD donors at 85.9% to 87.6% (Figure 4A). In contrast, the utilization rate of DBD organs from NID was only 43.4% in Era 1 and increased minimally to 43.6% in Era 3. Surprisingly, NID undergoing DCD donation were utilized more frequently than DBD kidneys from NID donors, with rates of 54.1% to 62.5% over the study period. The likelihood of NID + DBD donor utilization for kidney transplant decreased across eras (Table 6). NID + DBD donors were less likely to be used for kidney transplant in Era 2 (OR 0.882, 95% 0.822–0.946) and Era 3 (OR 0.938, 95% CI 0.876–01.004, *p* = 0.002) compared to Era 1. There was, however, no difference in donor utilization for kidney grafts from NID + DCD donors over time.

### 3.6. Organ-Specific Donor Utilization—Liver

The liver was the most frequently utilized organ from NID, being transplanted from 66.8% to 69.8% of donors over the study (*p* < 0.001). The utilization rate of DBD liver allografts from NID ranged from 70% in Era 1 to 76.7% in Era 3 but remained lower than DBD livers from ID which ranged from 85% to 86% (Figure 4B). The utilization rate for liver allografts from DCD donors dropped for ID from 41.3% in Era 1 to 28.4% in Era 3; as well as for NID, from 22% to 17.8% over the same time (*p* < 0.001). NID were more likely to have DBD liver utilization in both Era 2 (OR 1.211, 95% CI 1.131–1.297) and Era 3 (OR 1.511, 95% CI 1.411–1.618, *p* < 0.001) compared to Era 1 (Table 6). In contrast, compared to Era 1, NID were significantly less likely to be utilized for DCD liver allografts in Era 2 (OR 0.441, 95% CI 0.319–0.609) and Era 3 (OR 0.752, 95% CI 0.583–0.968, *p* < 0.001).

### 3.7. Organ-Specific Donor Utilization—Heart

Heart allografts were primarily obtained from DBD donors during the study period. There was a small but statistically significant increase in NID utilization for heart allografts from 4.0% in Era 1 to 5.4% in Era 2 and 7.1% in Era 3 (*p* = 0.006, Figure 4C). NID were significantly more likely to be used for heart transplant in Era 3 (OR 1.623, 95% CI 1.415–1.862, *p* < 0.001) compared to Era 1. No DCD organs were used for heart transplantation from NID during the study period.

### 3.8. Organ-Specific Donor Utilization—Lung

NID utilization for lung allografts increased substantially from 5.8% in Era 1 to 9.3% in Era 2, and 10.5% in Era 3 (*p* < 0.001, Figure 4D). DBD lungs from NID were progressively more likely to be utilized for lung transplant across eras (Table 6). Compared to Era 1, the likelihood of donor utilization in Era 2 was OR 1.615 (95% CI 1.436–1.818) and in Era 3 was OR 2.251 (95% CI 2.011–2.520, *p* < 0.001). Over the study period, the number of overall DCD lung donors increased but remained very low. Only 19 NID + DCD donors were used for lung transplant across the entire study period and therefore, additional analysis was not pursued.

## 4. Discussion

### 4.1. Definitions of Marginal and Non-Ideal Donors

The utilization of marginal donors has long been proposed to expand the donor pool and meet the persistent need for suitable organs for transplantation [24,25,26]. The present study created a broad and novel definition of non-ideal donor and evaluated the prevalence and utilization of NID as part of the donor pool.

To date, there is no existing generalized definition of a non-ideal donor for transplantation, making comparisons of donor utilization across organs difficult. For this study we generated a composite definition of non-ideal donors, derived from characteristics included in organ-specific definitions and seeking to include even young donors with multiple comorbid conditions. Prior definitions of “marginal” donors have been developed for each organ individually, including ECD definition for kidney donors, KDPI calculations, liver DRI, and lung Oto-Donor Score, amongst others (Table 1). Common factors across these scoring systems include donor age and medical comorbidities, such as hypertension and diabetes. To be both inclusive and discriminatory, the definition developed here included parameters based on donor comorbidities and varied with donor age, such that with increasing age, fewer comorbid conditions were required to be considered marginal. Notably, DBD and DCD cohorts were separated for analysis rather than defining all DCD donors as marginal. The purpose of this was to recognize that the potential organ injury associated with DCD procurement and organ preservation differs from organ quality associated with age and chronic illness.

This analysis is intended to identify potential donors at risk for organ non-utilization. This concept of a non-ideal donor may be most useful to OPOs attempting to evaluate an ever increasing number of potential donors. Early identification of donors at low probability for multi-organ donation based on DUS may save futile organ offers, expedite donors to procurement, and minimize risk for donor decompensation during organ allocation. By providing a metric to prioritize donors at highest likelihood for multi-organ donation, efforts can be focused to maximize organ yield from appropriate donors. The concept of a non-ideal donor may also help inform transplant centers, who often delay on-site preparations pending the unknown disposition of multivisceral offers. DUS may help centers predict likelihood of multivisceral transplant, mobilize potential recipients, and minimize cold ischemic time.

Furthermore, this definition will prove useful when attempting to understand the effects of machine perfusion technology on organ utilization. Identification of non-ideal donors allows for a standardization of global donor factors when assessing organ-specific changes in utilization. These results may also inform which systemic donor factors should be considered indications for use of machine perfusion and advanced organ preservation technologies. Future directions using the concept of non-ideal donor and DUS include analysis of machine perfusion utilization rates based on systemic donor factors, as well as analyzing organ-specific utilization and graft outcomes based on DUS. Overall, the concept of a non-ideal donor may help us understand the impact of increasing comorbidities in the donor population as they are reflected in transplant outcomes.

It is also important to note that several donor factors considered to be “high risk” or “marginal” in nature do not correlate with organ utilization or outcomes. For example, although a history of drug use identifies donors as “Public Health Service (PHS) High Risk”, our group has previously demonstrated that these donors tend to be younger overall and have increased liver utilization from DBD donors as compared to other causes of death [27]. In addition to increased liver utilization, liver allografts from donors deceased of drug intoxication also demonstrate superior one-year graft survival over other causes of death [28]. Through validation of the DUS as it relates to organ utilization, we have identified a set of global donor comorbidities that correlates to the number of potential donors and overall organ utilization.

### 4.2. Organ-Specific Donor Utilization

Importantly, this study demonstrates that the utilization of non-ideal donors is increasing over time, but is variable with donor neurologic status. Compared to the earliest era (2005–2009), non-ideal DBD donors were 20% more likely to be utilized in 2010–2014 and 40% more likely to be utilized in 2015–2019. This likely reflects the changing attitudes towards NID utilization across the transplant community. A significantly smaller proportion of DCD donors were classified as NID; however, the proportion of NID + DCD donors increased throughout the study period with significant increases between 2014 and 2019. This trend is consistent with the general trend in the United States, which reflects an overall increase in the number of DCD donors over the same time period [29]. It is important to note that DCD donor utilization primarily reflects liver and kidney utilization, compared to DBD donors, which may be used for up to five organs. The lower utilization rates amongst non-ideal DCD donors likely reflects a lower threshold for acceptable organ quality, despite increases in utilization of machine perfusion and advanced organ preservation technology in DCD heart and lung transplantation.

An interesting finding in this study is the low utilization rate of kidney allografts from NID. Kidney utilization rates for ID were comparable between DBD and DCD donors, between 84% and 86%. In contrast, the utilization of DCD kidneys from NID was higher than that for DBD kidneys from NID, and both were significantly lower than ID. The low rate of donor utilization from NID reflects prior studies showing increasing discard rates for NID kidneys. Stewart et al. found that kidney discard was associated with increasing donor age, elevated KDRI, HCV status as well as increasing utilization of donor biopsy—all factors which are part of an expanded donor pool [30]. The present study would suggest that the expanded donor pool remains under-utilized, with only 45% of donors being utilized for transplant and a 15% decreased likelihood of utilization in recent years. The implications of kidney discard and donor non-utilization cannot be overstated. Husain et al. showed that up to 30% of waitlisted candidates who declined a deceased donor kidney offer died or were removed from the waiting list before transplantation [31]. Successful utilization and transplantation of marginal organs relies on appropriate donor-recipient matching. Multiple studies have showed an overall survival benefit from transplantation with “marginal” donors compared to remaining waitlisted for nearly all candidates [32,33,34]. The results of the present study emphasize the need to maximize kidney utilization from NID, considering that the donor population is growing increasingly older, more obese, and has a higher prevalence of comorbid conditions [35].

In contrast, there was increased utilization of NID for all other organs. DBD donors were 47% more likely to be utilized for liver transplantation from NID in 2015–2019 compared to 2005–2009. Although less donors were evaluated for liver donation with rising DUS, organ utilization rates remained consistently high at 67–82%. This expansion of liver utilization is consistent with previous findings demonstrating satisfactory transplant outcomes from high-DRI donors, DCD donors, and liver donors with advanced age [36,37]. While age was included in our definition of non-ideal donors, we also considered other donor comorbidities. Takagi et al. recently performed a systematic review of 11 articles, finding donor BMI influenced liver utilization but not transplant outcomes and presumed poor quality livers were discarded based on biopsy finding [38]. Future observation of trends in liver transplant may make BMI increasingly less relevant, with increased adoption of machine perfusion offsetting some risks associated with graft steatosis and donor BMI. In contrast to age and BMI, donor diabetes has been associated with increased rates of primary non-function and reduced graft survival [39,40], and remains relevant to NID utilization. Utilization of livers from HCV-positive donors is increasingly common in the current age direct-acting antiviral therapy, with demonstration sustained virologic response and good graft survival [41,42,43]. Although HCV status did not seem to significantly impact utilization of liver allografts from NID, it did correlate with acceptance practices of other organs. As comfort with HCV-positive organs improves, HCV status will likely play a diminished role in organ utilization in the near future.

Approximately 15–25% of donors in the total donor pool meet “standard criteria” for lung transplant [44]. Here, we demonstrate an increase of >50% lung utilization from ID, as well as a two-fold increase in NID lung utilization between 2005–2009 and 2015–2019. Although not a focus of this study, others have demonstrated that use of expanded criteria donors for lung transplantation leads to comparable graft- and patient survival [45]. Some measures of extended criteria donors in lung transplant were not included in our definition, including smoking history and low PaO2; however, these measures are not alone contraindications to transplantation and organs with these characteristics have been used with acceptable outcomes [46,47,48]. Nonetheless, the factors included in the DUS correlated well with number of potential lung donors, as well as lung utilization rates.

In this study, utilization of heart allografts was primarily demonstrated from ID, with utilization approaching 50% in the 2015–2019 era. Although utilization of hearts from NID increased by 31%, this only reflects an absolute increase from 3.4% to 4.4%. Donor age remains a primary predictor in heart transplantation, particularly for recipients over 40 years old [49]. In Spain, utilization of donors >50 years old has been associated with similar graft survival as younger donors, but with increased risk of cardiac allograft vasculopathy [50]. Nevertheless, there may be a survival benefit with utilization of NID, although an increase in “marginal donor“ use for heart transplantation has been associated with increased risk of post-transplant morality [51]. Notably, cold and warm ischemia times were not included here, and impact graft outcomes beyond the characteristics inherent to the donor. This study does not capture the recent increase in utilization of heart allografts from DCD donors since 2019, employing both machine perfusion and normothermic regional perfusion to further increase heart utilization.

### 4.3. Future Directions

Beyond changes to donor utilization practices alone, there have been significant advancements in transplantation that will promote utilization of organs from NID. Recently, the use of normothermic regional perfusion or machine preservation has supported an increase in the recovery of hearts and lungs from DCD donors. While this currently represents a fraction of heart or lung transplants, adoption of these practices is already increasing donor utilization for these organs [52]. Furthermore, multiple studies have supported repair of lung allografts during ex vivo perfusion leading to successful transplantation and multiple ongoing trials (i.e., NOVEL, EXPAND, PERFUSIX trials) may enhance these results [53,54]. Utilization of machine perfusion for liver transplant is growing following the FDA approval of OCS Liver System (Transmedics, Andover, MA, USA) for normothermic liver preservation; however, multiple normothermic and hypothermic ex situ perfusion modalities are already broadly used in Europe and will likely expand to the U.S. Similarly, a new device for hypothermic oxygenated perfusion for kidney allografts has recently gained FDA approval (Kidney Assist Transport, XVIVO, Goteberg, Sweeden) and may further enhance the utilization of marginal renal allografts [55]. With these advances, we expect increasing utilization of non-ideal donors to meet the ever-present need for suitable allografts.

This present study provides a comprehensive overview of organ utilization in the United States over the past 15 years, however it is not without limitations. Importantly, our focus in this study was to evaluate organ utilization based on global donor comorbidities, and does not include the outcomes of transplanted grafts. Organ utilization and the outcomes of the subset of grafts selected for transplantation are both important and somewhat distinct issues that contribute to the overall shortfall of available organs for transplant. By focusing on organ allocation, we hope that this study prompts further consideration of donor qualities that inform acceptance practices. We intend for graft outcomes to be the focus of future study. The use of a large database is both a significant strength and weakness of this study. While providing a large amount of data over a broad time period, nuances which influence the utilization (or non-utilization) of a donor may be missed in the data recording and available variables.

Another limitation is the exclusion of certain factors which may relate to organ utilization. In particular, organ biopsy is used for livers and kidneys primarily to provide valuable information about organ quality and may significantly influence utilization [56,57,58]. Other notable factors which were not widely available and influence thoracic organ utilization include chest X-ray findings and PaO_2_/FiO_2_ ratios for potential lung donors, heart abnormalities identified by echocardiograms or coronary angiography (e.g., ejection fraction, left ventricular hypertrophy). Lastly, allocation policy has a direct effect on organ utilization; however, changes in allocation policies for each organ have occurred at various time points throughout the study period. Therefore, eras of equal duration were determined for evaluation of donor utilization. Nevertheless, a single donor may give rise to multiple organs and therefore overall donor utilization is subject to less confounding than organ-specific utilization throughout the study. A deeper analysis of factors affecting organ-specific utilization may inform donor management and organ allocation in the future to reduce discard and improve efficiency.

## 5. Conclusions

In an era where there is a persistent need for suitable donor organs and organ stewardship is of the utmost importance, the data presented here show progressive trends in donor utilization. These trends suggest two key points: firstly, as a transplant community, the U.S. is becoming more adept at successfully transplanting organs from non-ideal donors; and secondly, donor service areas should continue to pursue non-ideal donors as acceptable donors for transplantation. With developments in DCD donor management and advances in machine preservation and post-procurement organ optimization, there is room for increased aggressiveness and improvement to facilitate more transplants in the future.

## Figures and Tables

**Figure 1 jcm-13-03271-f001:**
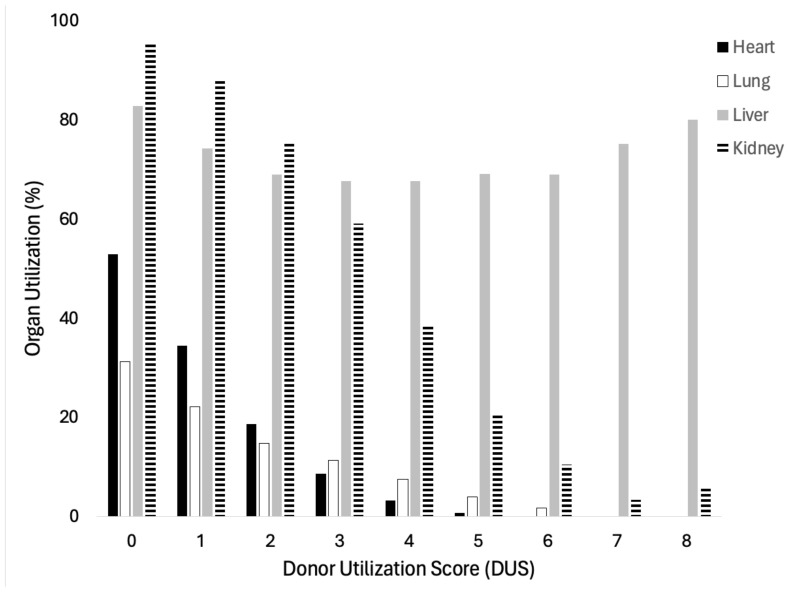
Organ-specific analysis of Donor Utilization Score versus organ utilization. Increasing DUS correlates with decreasing utilization of heart, lung, and kidney allografts.

**Figure 2 jcm-13-03271-f002:**
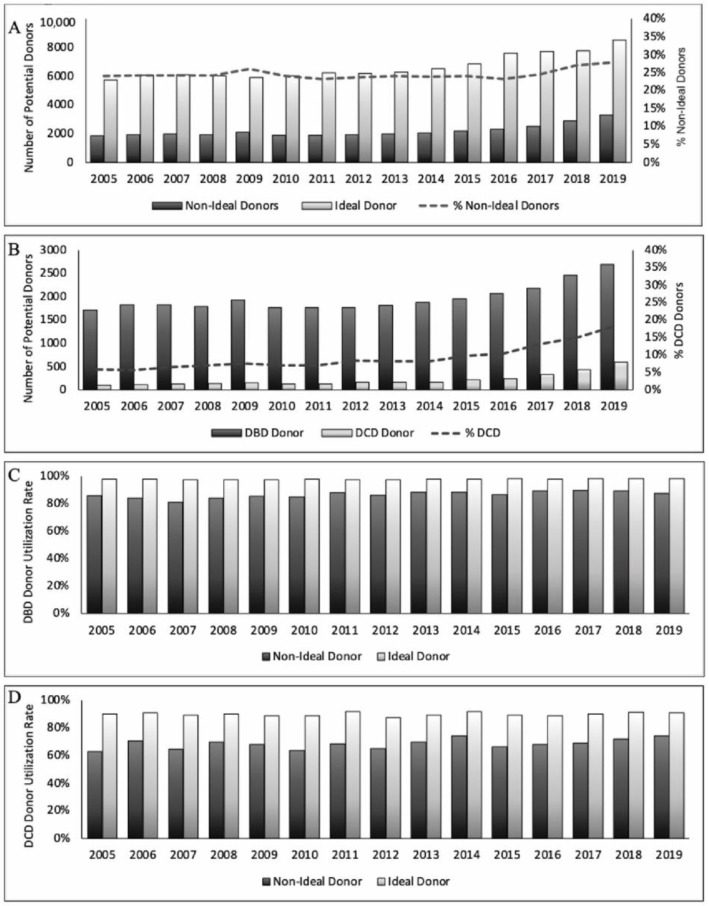
Distribution of non-ideal donors (NID) from 2005 to 2019. (**A**) Absolute numbers of ID and NID increased over time (*p* < 0.001 by Cochran-Armitage trend test) with proportion of NID donors shown (dotted line). (**B**) Distribution of donation after brain death (DBD) and donation after circulatory death (DCD) amongst NID with increasing proportion of DCD donors (*p* < 0.001 by Cochran-Armitage trend test). Utilization rates of ID and NID amongst DBD (**C**) and DCD (**D**) donors.

**Figure 3 jcm-13-03271-f003:**
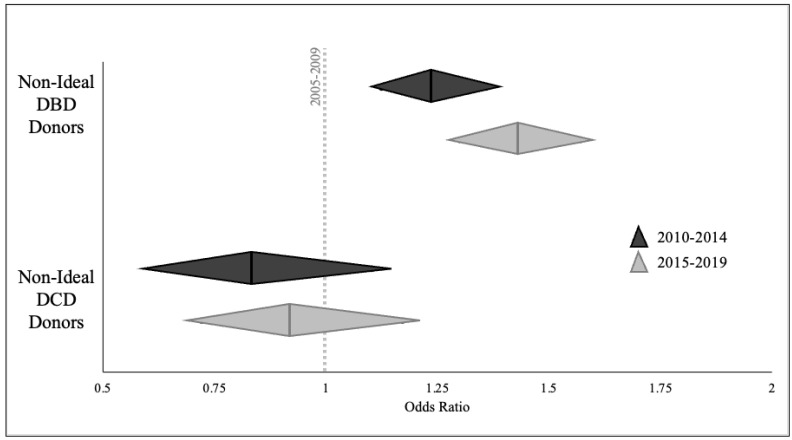
Likelihood of non-ideal donor (NID) utilization across eras. Multivariable models were created to evaluate likelihood of NID utilization and compared to the reference era (2005–2009, dotted line at OR = 1.00). Diamonds identify odds ratio (broad middle) and 95% confidence intervals (lateral points) of donor utilization in 2010–2014 (black) and 2015–2019 (grey). Donation after brain death (DBD) and donation after circulatory death (DCD) donors were evaluated separately.

**Figure 4 jcm-13-03271-f004:**
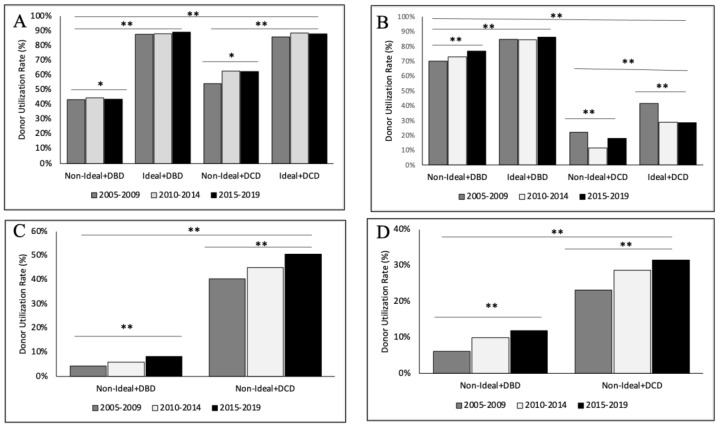
Donor utilization rates of Ideal and Non-Ideal Donors undergoing both DBD and DCD donation for (**A**) Kidney, (**B**) Liver, (**C**) Heart and (**D**) Lung allografts. Donation rates were compared across donor types and across donation eras. * For *p* < 0.01, ** for *p* < 0.001.

**Table 1 jcm-13-03271-t001:** Organ-Specific Donor Quality and Risk Scores.

	Scoring System	Variables Evaluated
Kidney	Extended Criteria Donor (ECD) [14]	Age, history of HTN, cerebrovascular accident as COD, terminal SCr >1.5 mg/dL
	Kidney Donor Risk Index (KDRI) [2]	Age, ethnicity, SCr, HTN, DM, COD, height, DCD, weight, donor type, HCV status, HLA mismatch, CIT, *en bloc* or double kidney transplant
Liver	Donor Risk Index (DRI) [4]	Age, COD, race, DCD, whole or split graft, height, CIT, location of organs
	Eurotransplant-DRI [15]	Age, COD, serum GGT, DCD, whole or split graft, rescue allocation, CIT
	Donor Quality Index (DQI) [16]	Age, COD, ICU stay, lowest MDRD creatinine clearance, whole or split graft
Lung	Oto-Donor Score [17]	Age, smoking history, CXR findings, secretions, PaO_2_/FiO_2_
	Eurotransplant Score [18]	Age, general and smoking history, CXR findings, bronchoscopy findings, PaO_2_/FiO_2_ ratio
	Minnesota-Donor-Lung Quality Index [19]	Donor age, recipient age, CIT, preexisting lung disease, smoking history, ABG values, procurement complexity, HLA matching, size mismatch, lung allocation score; risk of pneumonia, aspiration, pulmonary edema, pulmonary malignancy, donor transmitted disease, extrapulmonary malignancy and contusions
	Maryland-UNOS-data donor score [20]	Age, DM, smoking history, race
	Louisville-UNOS-data donor score [21]	Age, DM, smoking history, race
	Zurich Donor Score [22]	Age, smoking history, DM, significant pulmonary infection, PaO_2_/FiO_2_
Heart	Heart Donor Score [23]	Age, COD, compromised history, HTN, cardiac arrest, LVEF, valve function, LVH, coronary angiography, serum sodium, vasopressor support (e.g., norepinephrine, dopamine, dobutamine)
	Heart Donor Risk Index [10]	CIT, age, race, blood urea nitrogen/creatinine ratio

Abbreviations: ABG—arterial blood gas, CIT—cold ischemic time, COD—cause of death, CXR—chest X-ray, DCD—donation after circulatory death, DM—diabetes mellitus, GGT—gamma glutamyl transferase, HCV—hepatitis C virus, HLA—human leukocyte antigen, LVEF—left ventricular ejection fraction, LVH—left ventricular hypertrophy, MDRD—modification of diet in renal disease, SCr—serum creatinine.

**Table 2 jcm-13-03271-t002:** Donor Utilization Score.

Donor Utilization Score (DUS)	
Variable	Point(s)
Age	
>70 years	3
50–69 years	1
<50 years	0
BMI > 30	1
Diabetes	1
Hypertension	1
Prior MI	1
HCV-positive	1
Terminal SCr > 1.5	1

**Table 3 jcm-13-03271-t003:** Donor Utilization Score and definition of Non-Ideal Donor. Multi-organ donor was defined as the likelihood to donate 2 or more solid organs. Mean number of organs utilized per donor displayed with standard deviation. Non-Ideal Donor (NID) was defined as any donor with a DUS ≥ 3.

DUS	Total Donors	Any Organs Donated	Multi-Organ Donor	Organs Donated (Mean ± SD)
0	41,769	41,310 (98.9%)	35,981 (86.2%)	4.17 ± 1.66
1	33,980	32,859 (96.7%)	25,025 (73.7%)	3.26 ± 1.57
2	24,503	22,764 (92.9%)	14,578 (59.5%)	2.54 ± 1.43
3	16,872	14,966 (88.7%)	7654 (45.4%)	2.02 ± 1.32
4	9372	7779 (83.0%)	2621 (28.0%)	1.49 ± 1.15
5	4023	3118 (77.5%)	586 (14.6%)	1.10 ± 0.92
6	1062	784 (73.8%)	83 (7.8%)	0.90 ± 0.72
7	279	214 (76.7%)	5 (1.8%)	0.80 ± 0.50
8	26	21 (80.8%)	0 (0.0%)	0.81 ± 0.40

**Table 4 jcm-13-03271-t004:** Characteristics of Ideal and Non-Ideal Donors.

Donor Characteristics	All Donors (n = 132,465)	Non-Ideal Donors (n = 32,710)	Ideal Donors (n = 99,755)	*p* Values
Age, years (median, IQR)	42 (26–54)	56 (51–64)	35 (22–48)	<0.001
Gender, female (n, %)	53,475 (40.4%)	14,484 (44.3%)	38,991 (39.1%)	<0.001
BMI, kg/m^2^	26.5 (22.8–31.1)	31.4 (26.7–35.9)	25.3 (22.1–29.0)	<0.001
BMI ≥30 kg/m^2^	39,415 (29.8%)	19,841 (60.7%)	19,574 (19.6%)	<0.001
Ethnicity				<0.001
White	88,010 (66.4%)	20,497 (62.7%)	67,513 (67.7%)	
Black	21,767 (16.4%)	7216 (22.1%)	14,551 (14.6%)	
Hispanic	18,474 (14.0%)	4022 (12.3%)	14,452 (14.5%)	
Asian	3200 (2.4%)	805 (2.5%)	2395 (2.4%)	
Other	1014 (0.8%)	170 (0.5%)	844 (0.9%)	
Blood Type				<0.001
O	63,511 (48.0%)	15,941 (48.7%)	47,570 (47.7%)	
A	48,760 (36.8%)	11,826 (36.2%)	36,934 (37.0%)	
B	15,658 (11.8%)	3914 (12.0%)	11,744 (11.8%)	
AB	4529 (3.4%)	1028 (3.1%)	3501 (3.5%)	
DCD Donor	19,248 (14.5%)	3214 (9.8%)	16,034 (16.1%)	<0.001
Cause of Death				<0.001
Anoxia	41,477 (31.3%)	9973 (30.5%)	31,504 (31.6%)	
Trauma	42,103 (31.8%)	3580 (10.9%)	38,523 (38.6%)	
CVA/Stroke	44,822 (33.8%)	18,495 (56.5%)	26,327 (26.4%)	
Other	4063 (3.1%)	662 (2.0%)	3401 (3.4%)	
Hx of Hypertension	46,501 (35.3%)	28,704 (87.8%)	17,797 (17.8%)	<0.001
Hx of Diabetes	15,558 (11.8%)	13,141 (40.2%)	2417 (2.4%)	<0.001
Prior MI	5162 (3.9%)	4198 (12.8%)	964 (1.0%)	<0.001
HCV-Positive	7463 (5.6%)	2804 (8.6%)	4659 (4.7%)	<0.001
Heavy Alcohol Use	22,943 (17.3%)	5482 (16.8%)	17,461 (17.5%)	0.002
Cigarette Smoker ^a^	31,687 (23.9%)	11,242 (34.4%)	20,445 (20.5%)	<0.001
Any Drug Use	52,686 (39.8%)	9578 (29.3%)	43,106 (43.2%)	<0.001
Serum Creatinine	1.0 (0.7–1.6)	1.7 (1.0–3.3)	1.0 (0.7–1.6)	<0.001
Serum AST	47 (28–91)	41 (26–81)	49 (29–95)	<0.001
Serum ALT	38 (22–76)	33 (20–62)	39 (23–80)	<0.001
Total Bilirubin	0.7 (0.4–1.1)	0.7 (0.4–1.1)	0.7 (0.4–1.1)	<0.001
INR	1.3 (1.1–1.4)	1.3 (1.1–1.4)	1.3 (1.1–1.4)	0.071
Serum Sodium	147 (142–153)	147 (142–153)	147 (141–153)	0.101

Abbreviations: ALT—alanine aminotransferase; AST—aspartate aminotransferase; BMI—body mass index; CVA—cerebrovascular disease; DCD—donation after circulatory death; HCV—hepatitis C virus; Hx—history; INR—international normalized ratio; MI—myocardial infarction. ^a^ History of smoking >20 pack-years.

**Table 5 jcm-13-03271-t005:** Organ-Specific Donation Rates for Non-Ideal Donors Across Eras.

	2005–2009 (n = 8562)	2010–2014 (n = 8398)	2015–2019 (10,894)	*p*-Values
Organ Donor (n, %)	8093 (83.0%)	8367 (85.8%)	11,383 (86.2%)	<0.001
Multi-Organ Donor	3185 (32.7%)	3441 (35.3%)	4827 (36.6%)	<0.001
Organs per Donor (median, IQR)	1 (1–3)	1 (1–3)	1 (1–3)	<0.001
Kidneys				<0.001
Transplanted	4302 (44.1%)	4482 (46.0%)	6104 (46.2%)	
Recovered—Not Transplanted	2933 (30.1%)	2858 (29.3%)	4183 (31.7%)	
Not Recovered	2516 (25.8%)	2415 (24.8%)	2917 (22.1%)	
Liver				<0.001
Transplanted	6517 (66.8%)	6657 (68.2%)	9059 (69.8%)	
Recovered—Not Transplanted	1548 (15.9%)	1302 (13.4%)	1434 (18.9%)	
Not Recovered	1686 (17.3%)	1796 (18.4%)	12,711 (20.5%)	
Heart				0.006
Transplanted	390 (4.0%)	523 (5.4%)	936 (7.1%)	
Recovered—Not Transplanted	11 (0.1%)	7 (0.1%)	15 (0.1%)	
Not Recovered	9350 (95.9%)	9225 (94.6%)	12,253 (92.8%)	
Lungs				<0.001
Transplanted	569 (5.8%)	902 (9.3%)	1392 (10.5%)	
Recovered—Not Transplanted	11 (0.1%)	33 (0.3%)	101 (0.8%)	
Not Recovered	9171 (94.1%)	8820 (90.4%)	11,711 (88.7%)	

**Table 6 jcm-13-03271-t006:** Likelihood of Non-Ideal Donor Utilization by Organ Across Eras.

	DBD Donors			DCD Donors		
	OR	95% CI	*p*-Value	OR	95% CI	*p*-Value
**Kidney**						
2005–2009	Reference		0.002	Reference		0.61
2010–2014	0.882	0.822–0.946		1.150	0.896–1.475	
2015–2019	0.938	0.876–1.004		1.176	0.949–1.458	
**Liver**						
2005–2009	Reference		<0.001	Reference		<0.001
2010–2014	1.211	1.131–1.297		0.441	0.319–0.609	
2015–2019	1.511	1.411–1.618		0.752	0.583–0.968	
**Heart**						
2005–2009	Reference		<0.001			
2010–2014	1.143	0.987–1.324				
2015–2019	1.623	1.415–1.862				
**Lung**						
2005–2009	Reference		<0.001			
2010–2014	1.615	1.436–1.818				
2015–2019	2.251	2.011–2.520				

Multivariable modeling included: Donor Era; Donor characteristics: age, gender, ethnicity, blood type, cause of death, history of diabetes, hypertension, prior myocardial infarction, hepatitis C virus status, hepatitis B virus status, alcohol use, cigarette smoker, drug use, serum Cr > 1.5, ALT > 150, INR > 1.5.

## Data Availability

No new data were created or analyzed in this study. Data sharing is not applicable to this article.

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
