# Peer review of "Organ Utilization Rates from Non-Ideal Donors for Solid Organ Transplant in the United States"

_jcm, 2024, doi:10.3390/jcm13113271_

Round 1
Reviewer 1 Report
Comments and Suggestions for Authors
This study aims at analyzing the use of solid organs from so-called marginal donors over time in the US.
For this purpose, the authors developed a new definition of a marginal donor that shall be applicable independent of the type of organ of the donors. According to the manuscript the definition of a marginal donor “was derived from current organ-specific definitions of extended criteria donors”. The authors follow the working principles of the Crystal City Meeting of 2001 published in AJT 2002 (this reference is actually missing in the reference list and was probably intended als reference 50, actually all references contained in table 1 are missing!). This approach is outdated, currently more sophisticated scoring systems have been developed. It is unclear, how the authors came to the selection of the comorbidities and the age groups used. The definition of an extended criteria/marginal organ should be based on clinical evidence showing that the use of such organs is linked with an increased risk for the success of the transplant like this is the case for most of the risk scores the authors list in table 1. (KDRI, DRI, Eurotransplant DR etc.). The authors fail to show this for their scoring system. In addition, it is unclear, why they aim for one common definition of a marginal donor organ to be used for all organs. It is with good reason that currently used risk scores vary from organ to organ – a history of a myocardial infarction obviously has a completely different impact on a successive heart transplantation than on a liver transplantation.
As a consequence, their definition of a marginal donor even leads to the paradoxical effect that the well-known factor for extended criteria organs (i.e. drug abuse) is more prominent in the standard donor group than in the marginal donor group.
There are several studies showing the change in the use of donor organs from “marginal” donors based on organ specific criteria – often with substantial differences between the US and other countries and over time. These organ specific reports are of higher relevance than the general, not organ-specific analysis presented here.
Interpretation of the data is further hampered by the fact that is unclear to what extend the different marginal donor subcategories (donor age > 70 years, donor age > 50 years and 2 of 6 comorbid conditions, age below 50 and 3 or more comorbid conditions) contribute to the group of marginal donors defined here and whether the different comorbid conditions had different impact on the acceptance rates for the different organs.
Many of the aspects that are discussed in the discussion section are in line with current knowledge and common opinion in the field of transplantation. Most are only loosely linked to the specific results of the data analysis provided and are therefore mor of an opinion document than a scientific analysis of the data provided.
A final major concern is the possibility that the definition of a “marginal donor” used in this publication could lead to the unintended effect that suitable organs are declined simply because they are labeled as “marginal organs” in a sort of self-fulfilling prophecy.
The value of the analysis could possibly be improved based on the available wealth of data by:
- Analyzing the utilization of standard and marginal organs for each type of organ based on established organ-specific risk scores.
- Refining the existing risk scores for the individual organs (ideally subdividing DBD and DCD donors) and analyze utilization rates based on these scores.
If the authors want to stick to the newly defined common definition of a marginal organ used in this paper they should show that the score is of equal relevance compared to the currently established scores.
Reviewer 2 Report
Comments and Suggestions for Authors
I read with interest the paper "increased utilization of marginal donors across solid organ transplant in the united states"
I found the paper interesting and findings well presented.
I have only a minor comment:
I suggest to split the discussion into subparagraph, for easier readability
Author Response
We appreciate this reviewer's favorable impression of our manuscript, and based on their comments, we have split the discussion section into sub-paragraphs. We agree this will improve the readability of the manuscript for the audience.
Reviewer 3 Report
Comments and Suggestions for Authors
Dear Authors,
I read with interest your article, evaluating the role of marginal donors in the transplant activity in the US.
I think the main issue is the definition of marginal graft. As my understanding, you create a single definition of marginal graft based on the current other index. Is this score validate by any other study? How do you deal with the possible bias given by an arbitrary inclusion/exclusion criteria? I think this has to be explained, because you are using a new definition of marginal donor that has not been proven before.
Also, what do you consider a standard donor? (Table 2)
Comments on the Quality of English Languageno major issue
Round 2
Reviewer 1 Report
Comments and Suggestions for Authors
Thanksnfor carefully addressing the concerns and suggestions . The manuscript has been substantially improved. While not everyone in the field will agree with the approach taken, the analysis could stimulate discussion on current policies and practices in organ donation and organ evaluation/acceptance strategies by OPOs and transplant centers.
Author Response
We appreciate your thoughtful feedback, which has improved the overall strength of the manuscript. As you suggest, we hope this will stimulate discussion as we continue to evaluate a higher number of donors with rising prevalence of comorbidities. Thank you-
Reviewer 3 Report
Comments and Suggestions for Authors
Dear Authors,
Thank you for your reply. The new score definition and results are clearer.
In addition, I would compare your score with the existing one to see if it can be used more widely.
Comments on the Quality of English LanguageNo major issue
Author Response
We sincerely appreciate the thoughtful feedback, as this has strengthened the manuscript.
With regards to comparison between our Donor Utilization Score and the Donor Risk Index, this will be a focus of future work. The DRI is predictive of graft outcomes, and we have not yet validated the Donor Utilization Score for outcomes of transplanted livers. A unified scoring system would be quite valuable overall, if indeed the same risk factors for non-utilization are also predictive of graft outcomes. We look forward to exploring this further in our future work, and appreciate your commentary.